# Food Safety Monitoring of *Salmonella* spp. in Northern Italy 2019–2021

**DOI:** 10.3390/pathogens12070963

**Published:** 2023-07-22

**Authors:** Daniela Manila Bianchi, Paola Barzanti, Daniela Adriano, Francesca Martucci, Monica Pitti, Carla Ferraris, Irene Floris, Roberta La Brasca, Carmela Ligotti, Sara Morello, Giulia Scardino, Noemi Musolino, Clara Tramuta, Cristiana Maurella, Lucia Decastelli

**Affiliations:** 1SC Sicurezza e Qualità degli Alimenti, Istituto Zooprofilattico Sperimentale del Piemonte, Liguria e Valle d’Aosta, Via Bologna 148, 10154 Turin, Italy; daniela.adriano@izsto.it (D.A.); francesca.martucci@izsto.it (F.M.); monica.pitti@izsto.it (M.P.); carla.ferraris@izsto.it (C.F.); irene.floris@izsto.it (I.F.); roberta.labrasca@izsto.it (R.L.B.); carmela.ligotti@izsto.it (C.L.); sara.morello@izsto.it (S.M.); giulia.scardino@izsto.it (G.S.); noemi.musolino@izsto.it (N.M.); clara.tramuta@izsto.it (C.T.); lucia.decastelli@izsto.it (L.D.); 2SC Epidemiologia e Analisi del Rischio, Istituto Zooprofilattico Sperimentale del Piemonte, Liguria e Valle d’Aosta, Via Bologna 148, 10154 Turin, Italy; paola.barzanti@izsto.it (P.B.); cristiana.maurella@izsto.it (C.M.); 3Centro di Riferimento per la Tipizzazione delle Salmonelle (CeRTiS), Istituto Zooprofilattico Sperimentale del Piemonte, Liguria e Valle d’Aosta, Via Bologna 148, 10154 Turin, Italy

**Keywords:** *Salmonella* spp., food safety, official food control

## Abstract

Salmonella is the second most frequent bacterial pathogen involved in human gastrointestinal outbreaks in the European Union; it can enter the food-production chain from animal or environmental sources or from asymptomatic food operators. European food legislation has established microbiological criteria to ensure consumer protection. *Salmonella* is listed under both process hygiene criteria and food safety criteria. Each EU member state designates an agency to organize or perform controls and other official activities. This paper describes the official control plans performed by competent authorities in Northern Italy in the three-year period 2019–2021. A total of 4413 food samples were delivered to the IZS Food Safety laboratories for *Salmonella* detection, of which 36 (0.8%) tested positive. *Salmonella* was most frequently detected in poultry meat samples (25/36 positive samples) followed by other meat products and pork products. The official controls for the protection of consumer health apply the EU’s farm-to-fork approach: the samples were collected during production (food production plants), from products on the market, and from collective catering (restaurants, cafeterias, canteens). This manuscript will provide information about the presence of *Salmonella* in foodstuffs that can help competent authorities to set control plans based on risk assessments.

## 1. Introduction

*Salmonella* is a Gram-negative, facultative anaerobe bacterium that inhabits the intestines of humans and animals [1]. Wild birds can be reservoirs as can domestic animals [2]. *Salmonella* can also be found in water, the environment, and contaminated food products [1].

The genus *Salmonella* is divided into two species: *S. enterica* and *S. bongori*. *S. enterica* has six subspecies: *enterica* (I), *salamae* (II), *arizonae* (IIIa), *diarizonae* (IIIb), *houtenae* (IV), and *indica* (VI) [3]. More than 2600 serovars of *S. enterica* are known [1]. *Salmonella* can grow in a solution of 0.4–4% sodium chloride and within temperatures from 5 to 47 °C (optimum range, 32–35 °C), is sensitive to heat (70 °C), and can grow in a pH from 4 to 9 (optimum range, 6.5–7.5 pH). While it can survive in dried food products, optimum water activity is between 0.99 and 0.94. *Salmonella* spp. is inhibited at pH < 3.8, temperature <7 °C, and water activity below 0.94 [1].

*Salmonella* is the second most frequent bacterial pathogen involved in human gastrointestinal outbreaks in the European Union, particularly *S.* Enteritidis and *S.* Typhimurium [4]. According to the European Food Safety Agency (EFSA) [5], salmonellosis was the second most often reported zoonosis in Europe in 2020 and the most frequently reported causative agent of foodborne outbreaks. Salmonellosis is characterized by gastroenteritis accompanied by nausea, vomiting, abdominal cramps, bloody diarrhea, headache, feverish conditions, and myalgia. Infants and the elderly are at greater risk of dehydration. While generally a self-limiting illness, salmonellosis-related deaths have been recorded in the very young, the elderly, and the immunocompromised [6].

*Salmonella* strains can enter the food production chain from animal or environmental sources or from asymptomatic food operators. According to the annual report on zoonoses in the European Union (EU) published by the EFSA and the European Center for Disease Prevention and Control (ECDC), most of the foodborne outbreaks in 2021 (733, 19.3% of the total) were caused by *Salmonella*, mainly after the consumption of eggs and egg products, bakery products, and meats and products thereof [7]. Pork meat is frequently associated with cases of salmonellosis [8]. Furthermore, *Salmonella* serovars involved in human infection were *S.* Enteritidis (54.6%), *S.* Typhimurium (11.4%), monophasic *S.* Typhimurium (8.8%), *S.* Infantis (2.0%), and *S.* Derby (0.93%) [7].

*Salmonella enterica*, for example, is responsible for 1.4 million cases of foodborne salmonellosis annually in the United States alone. Infection can occur after eating undercooked meat, poultry and eggs, and contaminated ready-to-eat products [9]. Various kinds of food can be contaminated by *Salmonella*. In 2022, a foodborne outbreak caused by multidrug-resistant monophasic *S.* Typhimurium linked to chocolate products [10] involved 150 cases reported across ten European countries and predominantly affected young children [11]. Another salmonellosis outbreak in Europe was linked to sesame-based products imported from Syria [12].

European food legislation has established microbiological criteria to ensure consumer protection. *Salmonella* is listed under both process hygiene criteria (indicating acceptable functioning of food production) and food safety criteria, which define the acceptability of a single food product or a batch of food products entering the market [13]. As concerns food safety criteria, European Commission regulation EC 2073/2005 [13] requires the absence of *Salmonella* in 25 g or mL in five sampling units of a wide range of foods: meat, milk and milk products, eggs and egg products, live bivalve mollusks and live echinoderms, tunicates and gastropods, and ready-to-eat foods. The absence of *Salmonella* in food production areas tested under process hygiene criteria is determined on the carcasses of cattle, sheep, goats, horses, and pigs; the absence in 25 g of a pooled sample of neck skin is checked in poultry carcasses of broilers and turkeys.

Under European legislation on food safety criteria for official control, analyses for the detection of *Salmonella* and the identification of serovars are fundamental to determine compliance or non-compliance of a food product, and of an entire food batch, with regulations. In *Salmonella* spp. detection, a food batch is considered unsafe for human consumption. Regardless of virulence genes or the strain’s antimicrobial-resistant profile, action must be undertaken to protect consumer health (withdrawal, recall, destruction or heat treatment).

According to Regulation (EU) 625/2017 [14] each Member State designates an agency to organize or perform controls and other official activities. In Italy, food safety agencies designated by the Ministry of Health, regional health departments, and local health services are mandated to set and carry out annual regional food-safety plans [15]. Based on risk assessment studies, the plans state the number of samples to be collected and analyzed and by what criteria, the type of food matrix to be investigated, and the point in the food chain to be tested (under the responsibility of food business operators or on the market). Analytical tests are conducted at all stages of the production chain: the official controls for the protection of consumer health apply the EU’s farm-to-fork approach, and, therefore, the samples were collected during production (food production plants) and on the market (for sale to the final consumer and in business-to-business sales). In addition, a small percentage of samples was also collected from collective catering outlets (restaurants, cafeterias, canteens) to monitor good hygiene practices.

Food safety agencies can make use of official laboratories for accredited analysis of food samples. In Italy, the Istituti Zooprofilattici Sperimentali (IZS) form the national network of laboratories that provide scientific support and chemical and microbiological analyses for food safety agencies and other control bodies.

This article describes the work of the food safety laboratory of the IZS, Turin, and reports the results of *Salmonella* detection analysis of food products collected during official monitoring of the food chain of animal and vegetable origin for the three-year period 2019–2021. The data reported here refer to the epidemiological situation in northwestern Italy, which comprises the three regions of Piedmont, Liguria, and Valle d’Aosta. The official laboratory for food safety analytical control for this area is the Istituto Zooprofilattico Sperimentale del Piemonte Liguria and Valle d’Aosta, a public health agency under the Ministry of Health and accredited according to international standards for organizational procedures.

## 2. Materials and Methods

### 2.1. Sampling

Food samples were collected in the context of official food safety monitoring by local health services (Piedmont, Liguria, Valle d’Aosta). All samples were collected using sterile instruments according to instructions for obtaining samples and for transport to the laboratory, as described in international standard protocols [16]. The food samples were delivered to the IZS laboratories at the temperature reported on the label by food business operators for correct conservation of the particular foodstuff and analyzed within 24 h after arrival at the lab.

### 2.2. Rapid Screening Methods

Detection of *Salmonella* spp. was performed by rapid screening methods validated according to ISO 16140:2021 [17] and accredited (Table 1).

### 2.3. Microbiological Methods

Isolation and identification of *Salmonella* spp. were performed according to ISO 6579 [18] using the media described in international standard protocols (Table 2).

Five suspected colonies from each Petri dish were streaked on Columbia blood agar (CBA) (Becton & Dickinson, Franklin Lakes, NJ, USA) and incubated at 35 °C for 24 ± 2 h. One isolated colony from each CBA plate was analyzed with a MALDI Biotyper^®^ Sirius System mass spectrometer (Bruker Daltonik GmbH, Bremen, Germany) using the extended direct transfer (eDT) procedure. One loop of biomass was transferred to a MBT96 polished steel BC target plate (Bruker Daltonik GmbH) spot. The air-dried sample spot was overlaid with 1 µL of a formic acid water solution (70% *v*/*v*) and, after air-drying, with 1 µL of matrix HCCA (α-cyano-4-hydroxycinnamic acid) solution (Bruker Daltonik GmbH). Target plates of the samples were analyzed, and spectra were acquired in positive ion mode in the *m*/*z* range 2000–20,000. As an alternative, one isolated colony from each CBA plate was streaked on triple sugar iron agar (TSI) (Biolife Italiana S.r.l., Milan, Italy). Positive TSI colonies were confirmed by biochemical analysis using API 20E galleries (bioMérieux, Marcy l’Étoile, France).

### 2.4. Serotyping

The strains confirmed as being *S. enterica* were subcultured on Columbia blood agar (Becton & Dickinson) at 37 °C for 24 h and then serotyped according to the Kaufmann–White–Le Minor scheme [19] using O and H antisera (Statens Serum Institut, Artillerivej, Denmark).

### 2.5. Statistical Analysis

After cleaning and preparing the data, descriptive statistical analysis was conducted to obtain information about the distribution of *Salmonella* detection analyses in food matrices, criteria, and sampling sites monitored during the three-year period and to investigate for a possible seasonal effect. Prevalence and confidence intervals were calculated using Stata Special Edition, release 17.1 (StataCorp, College Station, TX, USA).

## 3. Results

### 3.1. Samples Delivered to the Laboratories

A total of 4413 food samples were delivered to the IZS food safety laboratories for *Salmonella* detection during the three-year period 2019–2021. Three main sampling plans were in operation: (1) official monitoring of national food products (*n* = 4222); (2) official monitoring of imported/exported food products (*n* = 126); (3) official monitoring of imported/exported food products after previous reports of non-compliance (*n* = 65).

In 2019, 1548 samples were collected and analyzed; in 2020, 1404 samples; and in 2021, 1461 samples. Table 3 presents the number of samples collected by year and region: Piedmont, 1222 in 2019, 1080 in 2020, and 1152 in 2021; Liguria, 302 in 2019, 304 in 2020, and 279 in 2021; Valle d’Aosta, 24 in 2019, 20 in 2020, and 30 in 2021 (Table 3).

The entire food production and serving chain undergoes official monitoring by a competent agency. Sampling is conducted on foodstuffs falling under the responsibility of food business operators (in production plants) or on foodstuffs for sale on the market or at collective catering outlets (Table 4).

Food matrices were collected according to the regional plans for food safety based on risk assessment studies of food categories (Table 5).

### 3.2. Detection of Salmonella spp.

Out of the total of 4413 food samples collected and analyzed, 36 (0.8%) tested positive for *Salmonella* spp. Out of the total of 3328 samples collected on the market, 33 (0.99%) tested positive and 3 out of the total of 1030 (0.29%) collected at a production plant tested positive.

Of the food samples collected during monitoring of national foodstuff products (*n* = 4222), 30 tested positive (30/4222; 0.71%). The rates of positive samples collected during official monitoring of imported/exported food products and imported/exported food products collected for testing after previous results were non-compliant were 1/126 (0.79%) and 5/65 (7.69%), respectively.

*Salmonella* spp. was most frequently detected in poultry meat samples (25/36 positive samples) followed by other meat products and pork products. The remaining *Salmonella* strains were found in fishery samples, a raw-milk cheese sample, and in bulk cereals from a third country (Table 6). The percentage of positive samples by year ranged between 0.45% (2019) and 1.64% (2020) (Table 7).

Non-poultry meat samples testing positive were pork meat; the positive dairy sample was a raw-milk cheese; fishery products testing positive were *Gallus provincialis* (*n* = 1) and *Mytilus galloprovincialis* (*n* = 1); and a sample of soy seeds in the cereals and seeds category was also found to be positive.

### 3.3. Salmonella Serovars

Table 8 presents the *Salmonella* serovars according to the Kauffman–White scheme. *Salmonella* Infantis was the most frequently isolated and identified serovar (*n* = 19), followed by *S*. Derby (*n* = 3), *S.* Enteritidis (*n* = 3), and the monophasic variant *S.* Typhimurium 4,5,12;i;-; (*n* = 3). *S*. Infantis and *S.* Enteritidis were most often isolated from poultry meat, while *S*. Derby and *S.* Typhimurium were most often isolated from pork products. Other serovars were identified in one sample; one strain isolated from a fish product (*Gallus provincialis*) was classified as group *O:11*; *H: e,h,x* and could not be assigned to a specific serovar.

### 3.4. Seasonality of Salmonella Detection

The multivariable analysis investigated for a possible seasonal effect. The only time covariate that significantly associated with the detection of *Salmonella* spp. was the year 2020 as the time of data collection and analysis. Table 9 presents the prevalence ratio (PR) of exposed (year 2020) versus non-exposed (year 2019). The covariate was statistically significant: 1 not included in the 95% confidence interval.

## 4. Discussion

According to the online EFSA dashboard [20], between 2017 and 2021, sampling units from five food categories from 37 countries (29 EU member states) were tested for *Salmonella* spp. The food category with the highest number of positive units was meat and meat products (38,853 positive units out of a total of 997,615 units tested during 2021), followed by egg and egg products (77 positive units out of a total of 14,817 units tested during 2021), and fish and fishery products (63 positive units out of a total of 15,259 units tested during 2021). The categories fruit, vegetables, and juices (7 positive units out of a total of 12,485 units tested) and milk and dairy products (28 positive units out of a total of 45,182 units tested) had the lowest number of positive units during 2021.

The high incidence of *S*. Infantis detected in poultry meat is shared by the 2019 report published by the National Reference Center for Salmonellosis. In 2019, *S*. Infantis was the most often detected serotype in *Gallus gallus* farms and 44% of *S*. Infantis strains were detected in broilers [21]. *S*. Infantis was massively reported for broiler matrices in the EU in 2019, from animals (36.3% of all serotyped isolates), and from other food matrices (49.1%). More than 50% of the *S*. Infantis isolates from broilers in 2019 were reported for Italy. In addition, in 2020, *S*. Infantis and *S*. Derby isolates were most often reported for Italy, which accounted for 43% and 38.3%, respectively, of the isolates positive for these serovars [5]. Broiler meat is a common source of *Salmonella*, and the contamination of broiler farms has been increasingly associated with persistent serovars, such as *S*. Infantis [7].

Considering human cases of nontyphoid salmonellosis in northwestern Italy, the most often detected serovars in the period 2017–2021 were the monophasic variant *S.* Typhimurium 4,5,12;i;- (45.5%), *S.* Typhimurium (13.8%), and *S.* Enteritidis (9.4%). The average proportion of human cases caused by *S.* Infantis during the same period was 1.7%, with a rise from 0.5% in 2017 to 2.5% in 2021 [22]. The increase in the proportion of human cases attributable to *S.* Infantis is consistent with the data reported for food samples and for broiler farms from the National Reference Center for Salmonellosis.

A certain seasonality for *Salmonella* detection has been suggested. Salmonellosis may be caused by the convergence of different factors, including human behavior, prevalence in animal reservoirs, consumption patterns, and bacterial environmental survival. Variation in seasonal prevalence seems to be greater during warmer months and lower during colder months [23]. The multivariable Poisson regression model we used to investigate for a time effect revealed that the covariates representing the season in which the sample was collected and analyzed were not statistically significant. The observation period was quite short and other variables related to the COVID-19 pandemic might have made it difficult to identify an expected seasonal trend. The only significant time-related covariate was the difference in rates recorded for 2020 and 2019: the PR reflects the high proportion of *Salmonella* spp. positive samples in 2020 versus 2019 (3.62 times higher). Nontyphoidal salmonellosis, such as *S.* Enteritidis, *S.* Newport, and *S.* Typhimurium serotypes, are frequently associated with foodborne disease outbreaks from contaminated eggs, meat, milk products, and poultry [24].

The safety monitoring plans for foods of animal and vegetable origin are carried out at the local level and based on the number of food-producing plants located in the area and the size of the resident population. The IZS covers all of northwestern Italy, which includes Piedmont (about 4.3 million inhabitants), Liguria (about 1.5 million inhabitants), and Valle d’Aosta (about 125,000 inhabitants). Considering the differences in regional population density, the number of samples collected per region appears to be consistent. Furthermore, the number of analyses carried out in the three regions was constant during the three-year period for the total number of samples collected and for the type of matrix delivered to the laboratory.

The IZS laboratories use rapid screening methods to detect food-related pathogens, with negative results obtained in less than 24 h from the start of analysis (less than 48 h after collection): this is essential in the monitoring of fresh and perishable food. If the lab test result is non-compliant with established legal limits, food health officers undertake procedures to withdraw and recall food products considered unsafe according to Regulation (EU) 178/2002, art. 14 [25].

Monitoring of food imports from third countries entails a series of random samplings. The three-region positivity rates for *Salmonella* spp. were similar to the national rates in official controls; however, following reports of non-compliance at the time of importation into Europe through an Italian border point, subsequent analytical checks are performed to protect consumer health in EU member states and to prevent entry of potentially harmful food products into the EU.

In the context of official monitoring, controls following previous non-compliance reports play a vital role in protecting consumer health. Detection of unhygienic practices at food plants or in relation to certain food products is followed up with subsequent checks, as confirmed by these data. The food safety agencies perform serial controls to investigate the source of contamination and to determine whether corrective actions implemented by food business operators are truly effective.

Our data show a higher percentage of *Salmonella* positivity in food samples recorded for 2020. Food safety monitoring and sampling continued to protect consumer health during the COVID-19 pandemic. The relatively higher percentage of positives is not statistically significant, and there are currently no published data to compare similar trends for other EU countries or other food pathogens. The economic losses and difficulties in operations management experienced by food business operators because of the pandemic restrictions may have affected food safety standards in supply chains and production plants. This issue is awaiting consideration by other working groups and laboratories involved in food safety monitoring.

## Figures and Tables

**Table 1 pathogens-12-00963-t001:** Rapid and alternative methods validated and accredited for *Salmonella* spp. detection in food.

Method	Approach	Manufacturer Certification
ELFA	Immunoenzymatic	VIDAS-AFNOR BIO 12/32-10/11
Real-time PCR	Molecular biology	Bio-Rad-IQ Check Prep-AFNOR BRD 07/6-07/04 Applied Biosystem-AFNOR ABI 29/01-09/07

**Table 2 pathogens-12-00963-t002:** Culture media and incubation time and temperature for *Salmonella* spp. isolation.

Medium		Incubation
BPW	Buffered peptone water	18 ± 2 h; 37 ± 1 °C
RVS	Rappaport–Vassiliadis soja broth	24 ± 3 h; 41.5 ± 1 °C
MKTTn	Muller–Kauffmann tetrathionate novobiocin	24 ± 3 h; 37 ± 1 °C
XLD	Xylose lysinedesoxycholate agar	24 ± 3 h; 37 ± 1 °C
BGA	Brilliant green agar	24 ± 3 h; 37 ± 1 °C

**Table 3 pathogens-12-00963-t003:** Number of food samples collected by year and region.

Regions	2019	2020	2021	Total
Piedmont	1222	1080	1152	3454
Liguria	302	304	279	885
Valle d’Aosta	24	20	30	74
Total	1548	1404	1461	4413

**Table 4 pathogens-12-00963-t004:** Number of samples collected per year from food production plants and food serving/catering outlets.

	2019	2020	2021	Total
Market	1192	1064	1072	3328
Food production	341	317	372	1030
Food serving/catering	15	23	17	55
Total	1548	1404	1461	4413

**Table 5 pathogens-12-00963-t005:** Number of samples collected by year and food category.

Food Matrix	2019	2020	2021	Total
Non-poultry meat *	427	373	421	1221
Dairy products	299	247	258	804
Fish and fish products	185	247	183	615
Vegetables and fruits	128	108	111	347
Meat products *	118	89	127	334
Mixed food	96	68	92	256
Poultry meat *	110	105	73	288
Confectionery	59	62	72	193
Cereals and seeds	46	36	49	131
Eggs and egg products	35	35	42	112
Spices	22	22	19	63
Pasta	23	12	14	49
Total	1548	1404	1461	4413

* Non-poultry meat and poultry meat are fresh, refrigerated, unprocessed meat products; a meat product is a multi-ingredient food product in which meat is the main ingredient.

**Table 6 pathogens-12-00963-t006:** Number and percentage of *Salmonella*-positive samples by food category and year.

Food Matrix	Total	2019	2020	2021	Total	%
Non-poultry meat	1221	4/427	1/373	2/421	7	0.57
Dairy products	804	0/299	1/247	0/258	1	0.12
Fish and fish products	615	2/185	0/247	0/183	2	0.33
Poultry meat	288	0/110	21/105	4/73	25	8.7
Cereals and seeds	131	1/46	0/36	0/49	1	0.77
Total	4413	7/1548	23/1404	6/1461	36	0.81

**Table 7 pathogens-12-00963-t007:** Number and percentage of *Salmonella*-positive samples by year.

Year	Total	Number	%
2019	1548	7	0.45
2020	1404	23	1.64
2021	1461	6	0.41
Total	4413	36	0.81

**Table 8 pathogens-12-00963-t008:** *Salmonella* serovars by food category.

Serovar	No.	Poultry Meat	Pork Meat	Dairy	Cereals	Fishery
*S.* Infantis	19	17	2			
*S.* Derby	3	1	2			
*S.* Enteritidis	3	3				
*S.* Typhimurium *	3		3			
*S.* Agona	1	1				
*S.* Anatum	1	1				
*S.* Brandenburg	1			1		
*S*. Bredeney	1	1				
*S.* Minnesota	1				1	
*S.* Rissen	1					1
*S.* Thompsons	1	1				
*S.* enterica **	1					1
Total	36	25	7	1	1	2

* 4,5,12;i;- ** O:11; H: *e,h,x.*

**Table 9 pathogens-12-00963-t009:** *Salmonella* seasonality and prevalence ratio of 2020 compared to 2019.

Time Risk Factor	Exposure Level	No.	Pos (%)	PR (95% CI)
Year	2020	1381	23 (1.64%)	3.62 (1.55–8.44)
	2019	1541	7 (0.45%)	1 (reference)

## Data Availability

Data Availability Statements are available in section “MDPI Research Data Policies” at https://www.mdpi.com/ethics (accessed on 11 July 2023).

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
