# Peer review of "Food Safety Monitoring of Salmonella spp. in Northern Italy 2019–2021"

_pathogens, 2023, doi:10.3390/pathogens12070963_

Round 1

Reviewer 1 Report

Thank you authors for the study. Overall, the study report Salmonella prevalence based on national food safety monitoring programme over a 3-year period. I would like to offer some comments below for consideration. 

Line 141: What are '1), 2), and 3)'? Please explain to the readers.

Table 5: what are differences between non-poultry meat and meat products?

Line 180: Table X?

Line 195: Noted S. Infantis was the most frequently isolated serovar. Is the same serovar seen in human cases in the same country? Please discuss public health implication of this serovar.

Line 234-251: these explanations of different sampling programme should be described in the methodology.

Author Response

see attached file, thank you

Reviewer 2 Report

The manuscript entitled “Food safety monitoring of Salmonella spp. in northern Italy

2019-2021” is poorly written and there is not much novelty. Please read the following comments carefully

·         The manuscript should be revised for English editing and grammar mistakes.

·         Introduction: Rephrase the aim of the work to be clear and better sound.

·         In this study, number of isolates was too small to publish as epidemiological data that represent the situation in Italy.

·         The manuscript covers only isolation of Salmonella spp. and serotyping. It is of not much use to readers. Therefore, I suggest the authors to characterize the isolates for their virulence and antimicrobial resistance.

·         The authors have isolated 36 Salmonella in various food matrices in three years’ time. I would suggest the authors to undertake genomic studies to establish any phylogenetic or clonal relationship between the isolates and indicate insights about epidemiologic information of these isolates.

·         In material and methods section, authors have mentioned about statistical analysis. However, there is no mention of statistical analysis in results section.

·         In Table 6 Mention 36 food item (dish) for all 36 isolates, at least the readers would be interested in this information.

·         Line 107: Expand the isolation method of ISO 16140: 2021

·         Line 107 and 108: What do you mean by accredited

·         Table 1: Not clear. Mention the primers and methods used

·         Add the company, city, and country of the used bacterial media and reagents

·         Line 146: Table x?

·         Lines 150-153: Rephrase

·         Line 158: non-italicize spp.

English very difficult to understand/incomprehensible

Author Response

Please, see attached file, thank you

Round 2

Reviewer 2 Report

None

Minor editing of the English language required